# Effects of Shell Thickness on Cross-Helicity Generation in Convection-Driven Spherical Dynamos

**Luis Silva** [iD], **Parag Gupta** [iD], **David MacTaggart** [iD] and **Radostin D. Simitev** *[iD]

School of Mathematics and Statistics, University of Glasgow, Glasgow G12 8QQ, UK; lacsilva@gmail.com (L.S.); p.gupta.1@research.gla.ac.uk (P.G.); david.mactaggart@glasgow.ac.uk (D.M.)
* Correspondence: Radostin.Simitev@glasgow.ac.uk

**Abstract:** The relative importance of the helicity and cross-helicity electromotive dynamo effects for self-sustained magnetic field generation by chaotic thermal convection in rotating spherical shells is investigated as a function of shell thickness. Two distinct branches of dynamo solutions are found to coexist in direct numerical simulations for shell aspect ratios between 0.25 and 0.6—a mean-field dipolar regime and a fluctuating dipolar regime. The properties characterising the coexisting dynamo attractors are compared and contrasted, including differences in temporal behaviour and spatial structures of both magnetic fields and rotating thermal convection. The helicity $\alpha$-effect and the cross-helicity $\gamma$-effect are found to be comparable in intensity within the fluctuating dipolar dynamo regime, where their ratio does not vary significantly with the shell thickness. In contrast, within the mean-field dipolar dynamo regime the helicity $\alpha$-effect dominates by approximately two orders of magnitude and becomes stronger with decreasing shell thickness.

**Keywords:** rotating thermal convection; convection-driven dynamos; numerical simulations; bistability; mean-field magnetohydrodynamics; spherical shells

## 1. Introduction

Thermal flows give rise to some of the most characteristic large-scale features of cosmic objects—their self-sustained magnetic fields [1,2]. For instance, the Sun and several of the planets in the Solar System display substantial magnetic fields [3,4]. The solar magnetic field drives solar activity and strongly affects planetary atmospheres [5,6]. Earth's field shields life from solar radiation [7]. Farther out, the gas giants, the ice giants, and the Jovian moons all have significant magnetic fields [8]. These fields are sustained by dynamo processes in the interiors or the atmospheres of their celestial hosts where vigorous convective motions of electrically conductive fluids generate large-scale electric currents [9–11]. The convective flows are driven primarily by thermal buoyancy forces due to thermonuclear fusion in stellar interiors and secular cooling in planetary interiors, respectively. Thermal convection in celestial bodies is highly turbulent in nature and, at the same time, strongly influenced both by rotation and by the self-generated magnetic fields. Considerable attention has therefore been devoted to this fascinating and important subject, and for topical reviews we refer to the papers by Busse and Simitev [12], Jones [13], Wicht and Sanchez [14] and references within.

Conceptually, dynamo generation of large-scale magnetic fields is understood on the basis of mean-field dynamo theory [15–17], a well-established theory of magnetohydrodynamic turbulence. A cornerstone of the theory is the turbulence modelling of the mean electromotive force—the sole source term arising in the Reynolds-averaged magnetic induction equation governing the evolution of the large-scale field, see Section 3.5 further below. The electromotive force is usually approximated by an expansion in terms of the mean field and its spatial derivatives where the expansion coefficients are known informally as "mean-field effects". The turbulent helicity effect, also called $\alpha$-effect (in this

work, when we refer to "helicity" without further qualification, we intend the helicity associated with the $\alpha$-effect—this shorthand should not be confused with other helicities, such as "magnetic helicity"), has been studied extensively in the research literature on mean-field dynamo theory, for example, see [16,18] and references therein. In contrast, the cross-helicity effect, also known as $\gamma$-effect [19], has been a subject to a rather small number of studies, for example, [20,21] and works cited therein. This is due to the currently prevailing treatment of turbulence where large-scale velocity is neglected because of the Galilean invariance of the momentum equation. However, such treatment leads to the neglect of the large-scale shear effects, which are, in fact, significant. For example, large-scale rotation is ubiquitous in astro/geophysical objects, for example, the Solar internal differential rotation is substantial and well measured [22,23] while numerical simulations suggest it is an essential ingredient of the dynamo process and likely to be responsible for the regular oscillations of convection-driven spherical dynamos [24,25]. Similarly, a number of studies of plane-parallel flows confirm that cross-helicity effects are not small compared to helicity effects [26,27]. Apart from its role in dynamo generation, cross-helicity is an important Solar observable. For instance, measurements of the cross-helicity component $\langle u_z b_z \rangle$ at the Solar surface are available from the Swedish 1-m Solar Telescope and can be used to calculate the magnetic eddy diffusivity of the quiet Sun by quasilinear mean-field theory [28].

Cross-helicity has not been explored in models of self-consistent dynamos driven by thermal convection in rotating spherical shells and this paper aims to contribute in this direction. The main goal of this work is to investigate the relative importance of the helicity and cross-helicity effects as a function of the thickness of the convective shell. Intuitive arguments suggest that the $\alpha$-effect is important in the case of the geodynamo and the cross-helicity effect is important in the case of the global solar dynamo. Indeed, the geodynamo operates in the relatively thick fluid outer code of the Earth where large-scale columnar structures are believed to develop. The coherent columnar structures are characterised by relatively large-scale vorticity and generate a strong helicity $\alpha$-effect. In contrast, the global solar dynamo operates in the thinner solar convection zone where columnar structures are thought difficult to maintain and so vorticity may have a less regular structure, thus increasing the relative importance of the cross-helicity effect. To assess this hypothesis, we present a set of dynamo simulations that differ mainly in their shell thickness aspect ratio $\eta = r_i/r_o$, see Figure 1, while other governing parameters are kept fixed. Along with estimates of the relative strength of the helicity and cross-helicity effects, we report on the mechanisms of electromotive force generation and its spatial distribution. Variation of shell thickness is also relevant to the case of the geodynamo as the inner core did not exist at the time of formation of the Earth, but nucleated sometime later in the geological history of the planet and continues to grow in size.

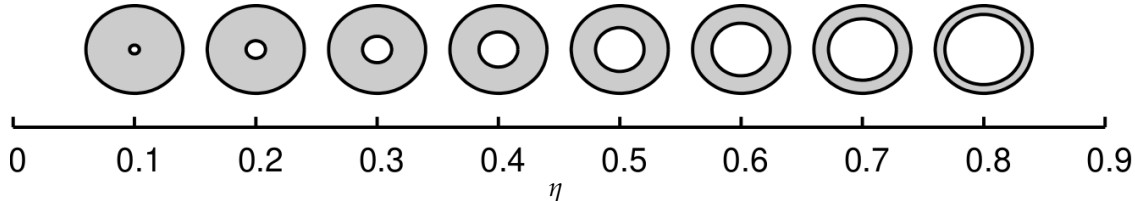

**Figure 1.** Illustration of shell thickness aspect ratio variation.

The geodynamo and the solar global dynamo are also different in that the former has a dominant and rarely reversing dipole, while the latter exhibits a regular periodic cycle. To capture this essential difference while comparing quid pro quo, we have performed this study at parameter values where two distinct dynamo branches are known to coexist [29–31]. These branches have rather different magnetic field properties, in particular one branch is non-reversing while the other branch is cyclic, and also display significant differences in zonal flow intensity and profile. It is reasonable to expect that the two branches will offer different mechanisms of helicity and cross-helicity generation and thus in this paper we proceed to study both branches. Bistability, in itself, may play a role in aperiodic magnetic

field polarity reversals, a notable feature of the geodynamo [32], as well as in the regular cycle of the solar dynamo [33]. We have previously investigated the hysteretic transitions between the coexisting dynamo branches with variation of the Rayleigh, Prandtl and Coriolis numbers (defined further below). In addition, in this paper we demonstrate for the first time that the distinct dynamo branches coexist also when the shell thickness $\eta$ is varied. The discussion of this dichotomous behaviour runs as a secondary theme of the article.

The paper is structured as follows. Details of the mathematical model and the numerical methods for solution are given in Section 2. In Section 3, we describe the set of dynamo simulations performed in the context of this work. We pay particular attention to the description of the two coexisting dynamo branches, which are studied for the first time here as a function of the thickness of the convective shell. In the process, we describe the typical morphology and time dependent behaviour of thermal convection flows. In Section 3.5, we briefly summarise the mean field arguments related to the helicity and cross-helicity mechanisms for the generation of a large-scale magnetic field. In Section 3.6, the cross-helicity properties of our dynamo solutions and the relative contributions of the $\alpha$-and $\gamma$-effects are assessed. Section 4 is devoted to concluding remarks.

## 2. Materials and Methods

This section describes a standard mathematical formulation of the problem of thermal convection and magnetic field generation in rotating spherical fluid shells. A set of transformations used to recast the problem in a scalar stream-function form and a pseudo-spectral algorithm used for the numerical solution of the equations is presented. The exposition in this section is standard and follows our previous articles, for example, [34,35]. This section also serves as an introduction and a review of the typical approach to the formulation and solution of this important problem.

### 2.1. Mathematical Formulation

We consider a spherical shell full of electrically conducting fluid as shown in Figure 2. The shell rotates with a constant angular velocity $\mathbf{\Omega}$ about the vertical coordinate axis. We assume that a static state exists with the temperature distribution

$$T_S = T_0 - \beta d^2 r^2 / 2, \tag{1a}$$

$$\beta = q / (3 \kappa c_p), \tag{1b}$$

$$T_0 = T_1 - (T_2 - T_1) / (1 - \eta). \tag{1c}$$

The evolution of the system is governed by the equations of momentum, heat and magnetic induction, along with solenoidality conditions for the velocity and magnetic fields,

$$\nabla \cdot \mathbf{u} = 0, \tag{2a}$$

$$(\partial_t + \mathbf{u} \cdot \nabla)\mathbf{u} = -\nabla \pi - \tau \mathbf{k} \times \mathbf{u} + \Theta \mathbf{r} + \nabla^2 \mathbf{u} + \mathbf{B} \cdot \nabla \mathbf{B}, \tag{2b}$$

$$P(\partial_t + \mathbf{u} \cdot \nabla)\Theta = R \mathbf{r} \cdot \mathbf{u} + \nabla^2 \Theta, \tag{2c}$$

$$\nabla \cdot \mathbf{B} = 0, \tag{2d}$$

$$P_m(\partial_t + \mathbf{u} \cdot \nabla)\mathbf{B} = P_m \mathbf{B} \cdot \nabla \mathbf{u} + \nabla^2 \mathbf{B}, \tag{2e}$$

written for the perturbations from the static reference state and with notations defined in Table 1. In this formulation, the Boussinesq approximation is used with the density $\varrho$ having a constant value $\varrho_0$ except in the gravity term where

$$\varrho = \varrho_0 (1 - \alpha \Theta), \tag{3}$$

and $\alpha$ is the specific thermal expansion coefficient $\alpha \equiv -(\mathrm{d}\varrho / \mathrm{d}T)/\varrho = \mathrm{const}$. With the units of Table 2, five dimensionless parameters appear in the governing equations, namely the shell radius

ratio $\eta$, the Rayleigh number $R$, the Coriolis number $\tau$, the Prandtl number $P$ and the magnetic Prandtl number $P_m$ defined by

$$\eta = \frac{r_i}{r_o}, \quad R = \frac{\alpha\gamma\beta d^6}{\nu\kappa}, \quad \tau = \frac{2\Omega d^2}{\nu}, \quad P = \frac{\nu}{\kappa}, \quad P_m = \frac{\nu}{\lambda}, \tag{4}$$

where $\lambda$ is the magnetic diffusivity. Since the velocity $\mathbf{u}$ and the magnetic flux density $\mathbf{B}$ are solenoidal vector fields, the general representation in terms of poloidal and toroidal components is used

$$\mathbf{u} = \nabla \times (\nabla v \times \mathbf{r}) + \nabla w \times \mathbf{r}, \tag{5a}$$

$$\mathbf{B} = \nabla \times (\nabla h \times \mathbf{r}) + \nabla g \times \mathbf{r}. \tag{5b}$$

Taking $\mathbf{r} \cdot \nabla\times$ and $\mathbf{r} \cdot \nabla \times \nabla\times$ of the momentum Equation (2b), two equations for $w$ and $v$ are obtained

$$[(\nabla^2 - \partial_t)\mathcal{L}_2 + \tau\partial_\varphi]w - \tau\mathcal{Q}v = \mathbf{r} \cdot \nabla \times (\mathbf{u} \cdot \nabla\mathbf{u} - \mathbf{B} \cdot \nabla\mathbf{B}), \tag{6a}$$

$$[(\nabla^2 - \partial_t)\mathcal{L}_2 + \tau\partial_\varphi]\nabla^2 v + \tau\mathcal{Q}w - \mathcal{L}_2\Theta = -\mathbf{r} \cdot \nabla \times [\nabla \times (\mathbf{u} \cdot \nabla\mathbf{u} - \mathbf{B} \cdot \nabla\mathbf{B})], \tag{6b}$$

where $\partial_\varphi$ denotes the partial derivative with respect to the angle $\varphi$ of a spherical system of coordinates $(r, \theta, \varphi)$ and where the operators $\mathcal{L}_2$ and $\mathcal{Q}$ are defined as

$$\mathcal{L}_2 \equiv -r^2\nabla^2 + \partial_r(r^2\partial_r),$$

$$\mathcal{Q} \equiv r\cos\theta\nabla^2 - (\mathcal{L}_2 + r\partial_r)(\cos\theta\partial_r - r^{-1}\sin\theta\partial_\theta).$$

The heat equation for the dimensionless deviation $\Theta$ from the static temperature distribution can be written in the form

$$\nabla^2\Theta + R\mathcal{L}_2 v = P(\partial_t + \mathbf{u} \cdot \nabla)\Theta, \tag{6c}$$

and the equations for $h$ and $g$ are obtained by taking $\mathbf{r}\cdot$ and $\mathbf{r} \cdot \nabla\times$ of the dynamo Equation (2e)

$$\nabla^2\mathcal{L}_2 h = P_m[\partial_t\mathcal{L}_2 h - \mathbf{r} \cdot \nabla \times (\mathbf{u} \times \mathbf{B})], \tag{6d}$$

$$\nabla^2\mathcal{L}_2 g = P_m[\partial_t\mathcal{L}_2 g - \mathbf{r} \cdot \nabla \times (\nabla \times (\mathbf{u} \times \mathbf{B}))]. \tag{6e}$$

For the flow we assume stress-free boundaries with fixed temperatures

$$v = \partial_{rr}^2 v = \partial_r(w/r) = \Theta = 0 \quad \text{at } r = r_i \text{ and } r = r_o. \tag{7a}$$

For the magnetic field we assume electrically insulating boundaries such that the poloidal function $h$ must be matched to the function $h^{(e)}$, which describes the potential fields outside the fluid shell

$$g = h - h^{(e)} = \partial_r(h - h^{(e)}) = 0 \quad \text{at } r = r_i \text{ and } r = r_o. \tag{7b}$$

This is a standard formulation of the spherical convection-driven dynamo problem [13,36–38] for which an extensive collection of results already exists [24,34,39,40]. The results reported below are not strongly model dependent as confirmed by simulations of convection driven by differential heating [41], for cases with no-slip conditions at the inner boundary and an electrical conductivity of the exterior equal to that of the fluid [25,42], and for thermo-compositional driving [35]. Thus, aiming to retain a general physical perspective, we intentionally use here a generic model formulation with a minimal number of physical parameters including only those of first-order importance for stellar and planetary applications.

**Table 1.** Notation used in Section 2.1, where not defined in the main text.

| Notation | Quantity | Notation | Quantity |
|---|---|---|---|
| $(r, \theta, \varphi)$ | Spherical polar coordinates | $T_S$ | Background temperature distribution |
| $t$ | Time | $T_1, T_2$ | Temperature inner, outer boundary |
| $\mathbf{r}$ | Position vector wrt centre of sphere | $q$ | Density of uniformly distributed heat sources |
| $d$ | Thickness of the spherical shell | $\kappa$ | Thermal diffusivity |
| $r_i, r_o$ | Inner and outer radii of the shell | $\nu$ | Kinematic viscosity |
| $\mathbf{u}$ | Velocity field perturbation | $\mu$ | Magnetic permeability |
| $\mathbf{B}$ | Magnetic flux density perturbation | $c_p$ | Specific heat at constant pressure |
| $\Theta$ | Temperature perturbation from the background state | $\gamma$ | Gravitational acceleration magnitude |
| $\pi$ | Effective pressure | $\partial$ | Partial derivative notation |

**Table 2.** Units of non-dimensionalisation.

| Quantity | Unit |
|---|---|
| Length | $d$ |
| Time | $d^2/\nu$ |
| Temperature | $\nu^2/\gamma\alpha d^4$ |
| Magnetic flux density | $\nu(\mu\varrho)^{1/2}/d$ |

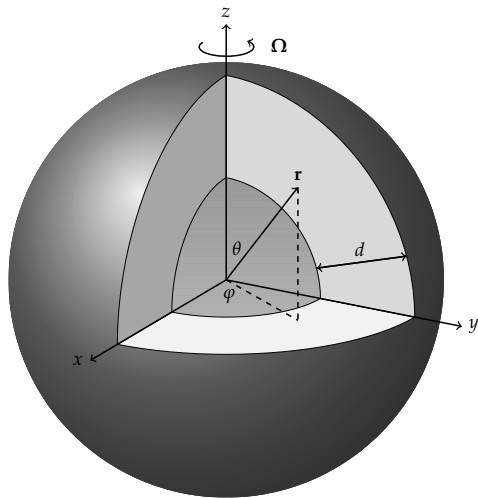

**Figure 2.** Schematic illustration of the three-dimensional region considered in this study, the associated spherical coordinate system and the position of the axis of rotation. The region is assumed full of electrically conducting fluid.

## 2.2. Numerical Methods

For the direct numerical integration of the convection-driven dynamo problem specified by the scalar Equation (6) and the boundary conditions (7) we use a pseudo-spectral method described by [43]. The code has been benchmarked for accuracy, most recently in [44,45], and has been made open source [46]. All dependent variables in the code are spatially discretised by means of spherical harmonics $Y_l^m$ and Chebychev polynomials $T_n$, for example,

$$v(r, \theta, \varphi) = \sum_{l,m,n}^{N_l, N_m, N_n} V_{l,n}^m(t) T_n \big(2(r - r_i) - 1\big) Y_l^m(\theta, \varphi), \tag{8}$$

and similarly for the other unknown scalars, $w$, $h$, $g$ and $\Theta$. The nonlinear terms in the equations are computed in physical space and then projected onto spectral space at every time step. Time integration makes use of an IMEX combination of the Crank–Nicolson scheme for the diffusion terms and the Adams–Bashforth scheme for the nonlinear terms with both schemes of second order accuracy.

When the spectral powers of the kinetic and magnetic energies drop by more than three orders of magnitude from the spectral maximum to the cut-off wavelength, we consider the simulations to be reasonably resolved [47]. In all the cases reported here, a minimum of 41 collocation points in the radial direction has been considered, together with spherical harmonics up to order 96. These numbers provide sufficient resolution, as demonstrated in Figure 7 for two typical dynamo solutions.

*2.3. Diagnostics*

It is convenient to characterise the non-magnetic convection and the convection-driven dynamo solutions using their energy densities. To understand the interactions between various components of the flow, we decompose the kinetic energy density into mean poloidal, mean toroidal, fluctuating poloidal and fluctuating toroidal parts as follows

$$\overline{E}_p = \frac{1}{2}\langle |\, \nabla \times (\nabla \overline{v} \times \mathbf{r})\,|^2 \rangle, \quad \overline{E}_t = \frac{1}{2}\langle |\, \nabla \overline{w} \times \mathbf{r}\,|^2 \rangle, \tag{9a}$$

$$\widetilde{E}_p = \frac{1}{2}\langle |\, \nabla \times (\nabla \widetilde{v} \times \mathbf{r})\,|^2 \rangle, \quad \widetilde{E}_t = \frac{1}{2}\langle |\, \nabla \widetilde{w} \times \mathbf{r}\,|^2 \rangle, \tag{9b}$$

where $\langle \cdot \rangle$ indicates the average over the fluid shell and time as described in Section 3.5 and $\overline{v}$ refers to the axisymmetric component of the poloidal scalar field $v$, while $\widetilde{v}$ is defined as $\widetilde{v} = v - \overline{v}$. The corresponding magnetic energy densities $\overline{M}_p$, $\overline{M}_t$, $\widetilde{M}_p$ and $\widetilde{M}_t$ are defined analogously with the scalar fields $h$ and $g$ for the magnetic field replacing $v$ and $w$.

To assess the predominant configuration of the magnetic field, we define the dipolarity ratio

$$\mathcal{D} = \overline{M}_p / \widetilde{M}_p. \tag{10}$$

When $\overline{M}_p > \widetilde{M}_p$ then $\mathcal{D} > 1$ and the corresponding solutions will be referred to as "Mean Dipolar", for reasons to be explained below, and denoted by **MD** following [29]. When $\overline{M}_p < \widetilde{M}_p$ then $\mathcal{D} < 1$ and the corresponding solutions will be referred to as "Fluctuating Dipolar" and denoted by **FD**.

To quantify heat transport by convection the Nusselt numbers at the inner and outer spherical boundaries $Nu_i$ and $Nu_o$ are used. These are defined by

$$Nu_i = 1 - \frac{P}{r_i R} \left.\frac{\mathrm{d}\overline{\overline{\Theta}}}{\mathrm{d}r}\right|_{r=r_i}, \qquad Nu_o = 1 - \frac{P}{r_o R} \left.\frac{\mathrm{d}\overline{\overline{\Theta}}}{\mathrm{d}r}\right|_{r=r_o}, \tag{11}$$

where the double bar indicates the average over the spherical surface.

Other quantities are defined in the text as required.

## 3. Results

*3.1. Parameter Values Used*

In order to investigate the effects of the shell thickness on the properties of non-magnetic convection and on dynamo solutions we perform a suite of numerical simulations varying the shell aspect ratio between $\eta = 0.1$ and $\eta = 0.7$. To compare the simulations on an equal footing, as well as to keep the number of runs required to a manageable level, all parameters except those depending on the aspect ratio are kept at fixed values. The value of the Prandtl number is set to $P = 0.75$ allowing us to use a relatively low value of the magnetic Prandtl number $P_m = 1.5$ as appropriate for natural dynamos. The Coriolis number is fixed to $\tau = 2 \times 10^4$ representing a compromise between the fast rotation rate appropriate for the geodynamo and the relatively slow rotation rate appropriate for the

solar dynamo. To ensure that dynamos are driven equally strongly, we fix the value of the Rayleigh number at 3.8 times the critical value $R_c$ for the onset of convection for each shell thickness aspect ratio as shown in Figure 3 below. The required values of the critical Rayleigh number are determined as explained in the next section where we also discuss general features of the onset of thermal convection.

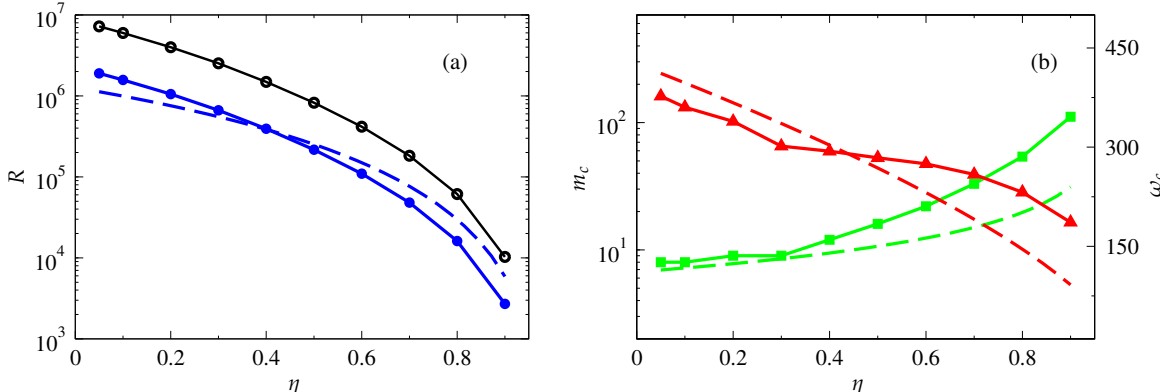

**Figure 3.** Critical parameter values for the onset of convection and values of the Rayleigh number used in this work as a function of the shell thickness aspect ratio $\eta$ in the case $P = 0.75$, and $\tau = 2 \times 10^4$. (**a**) The critical Rayleigh number $R_c$ for the linear onset of convection is plotted in solid blue curve marked by full circles. The values used in the simulations are given by $R = 3.8R_c$; they are plotted in solid black curve marked by empty circles. (**b**) The critical wave number $m_c$ (left *y*-axis) and the critical frequency $\omega_c$ (right *y*-axis) for the onset of convection are denoted by green squares and red triangles, respectively. Local asymptotic approximations (12) are shown by correspondingly coloured dashed curves in all panels. (Colour online).

### 3.2. Linear Onset of Thermal Convection

The onset of thermal convection in rapidly rotating spherical shells has been extensively studied, for example, most recently as a special case of the onset of thermo-compositional convection [48]. In general, two major regimes are found at onset–columnar convection and equatorially-attached convection. The equatorially-attached regime occurs at small values of the Prandtl number $P$ and consists of flows that take the form of non-spiralling rolls trapped near the equator with a relatively large azimuthal length scale. This regime can be understood as a form of inertial oscillations, for example, [49]. The columnar regime is realised at moderate and large values of $P$ and features elongated rolls parallel to axis of rotation that are spiralling strongly and have a relatively short azimuthal length scale. At the selected values of the Prandtl and the Coriolis numbers, the simulations reported in this study belong to the columnar regime of rapidly rotating convection.

To determine accurate values for the critical parameters at onset we use our open source numerical code [50]. The code implements a Galerkin spectral projection method due to Zhang and Busse [51] to solve the linearised versions of Equations (6a) and (6c). The method leads to a generalised eigenvalue problem for the critical Rayleigh number $R_c$ and frequency $\omega_c$ of the most unstable mode of thermal convection at specified other parameter values and at specified azimuthal wave number $m$ of the convective perturbation. Numerical extremisation and continuation problems then are tackled in order to follow the marginal stability curve in the parameter space as detailed in [48]. The critical values thus obtained are shown in Figure 3. The critical Rayleigh number $R_c$ and drift frequency $\omega_c$ decrease with decreasing shell thickness while the critical azimuthal wave number $m_c$ increases.

It is interesting to compare and validate these results against theoretical results for the onset convection in rapidly rotating systems. The asymptotic analysis of this problem has a long and distinguished history of local and global linear stability analysis [52–56], see also [48] for a brief

overview. Converting results of Yano [57] to our dimensionless parameters, length and time scales, we obtain

$$R_c = 7.252 \left( \frac{P\tau}{1+P} \right)^{4/3} (1-\eta)^{7/3}, \tag{12a}$$

$$m_c = 0.328 \left( \frac{P\tau}{1+P} \right)^{1/3} (1-\eta)^{-2/3}, \tag{12b}$$

$$\omega_c = 0.762 \left( \frac{\tau^2}{P(1+P)^2} \right)^{1/3} (1-\eta)^{2/3}, \tag{12c}$$

for the critical parameters of viscous columnar convection in an internally heated spherical shell. While expressions (12) are not strictly valid asymptotic results for the spherical shell configuration studied here, they provide a reasonable agreement with the numerical results plotted in Figure 3.

### 3.3. Finite-Amplitude Convection and Dynamo Features

As the value of the Rayleigh number is increased away from the onset, rotating columnar convection undergoes a sequence of transitions from steady flow patterns drifting with constant angular velocity to increasingly chaotic states as described in detail in [40]. When the amplitude of convection becomes sufficiently large so that the magnetic Reynolds number defined as $Rm = Pm\sqrt{2E}$ reaches values of the order $10^2$, onset of dynamo action is typically observed [34]. Three examples of dynamo solutions are shown in Figure 4 to (i) illustrate typical spatial features of chaotic thermal convection in rotating shells and the associated magnetic field morphology and (ii) to reveal how these features vary with decreasing shell thickness. Outside of the tangent cylinder the flow consists of pairs of adjacent spiralling convection columns as seen in the second row of Figure 4. Within the columns the fluid particles travel in clockwise and anticlockwise directions parallel to the equatorial plane and up towards the poles or down towards the equatorial plane as columns extend through the height of the convective shell. In agreement with the linear analysis, as the shell thickness is decreased the azimuthal wave number rapidly increases with the thin shell solution $\eta = 0.7$ showing a cartridge of fine scale columns closely adjacent to each other and exhibiting much weaker spiralling and slower drift than in the thick shell cases. These convective patterns strongly influence the structure and the morphology of magnetic fields as illustrated by the first row of Figure 4 where magnetic fieldlines of the three dynamo solutions are shown. The fieldlines are intricately knotted and exhibit a rather complicated structure within the convective domain in all three cases. The imprint of the convective columns is visible in the thick shell cases $\eta = 0.2$ and $\eta = 0.4$ where the magnetic fieldlines are coiled around the convective columnar structures indicating the presence of toroidal field and poloidal field feedback and amplification processes. Outside of the convective domain, the magnetic field of the thickest shell case $\eta = 0.2$ is well organised and emerges from the polar regions of the domain in the form of big bundles of opposite polarities with fieldlines proceeding to close and forming extensive overarching loops that are characteristic of a strong dipolar field symmetry. A similar picture is seen in the mid-thickness case $\eta = 0.4$ although in this case there appear to be several magnetic "poles" where strong bundles of vertical fieldlines emerge at the surface of the spherical domain. In the thin shell case $\eta = 0.7$ the magnetic field is much less organised with numerous fieldline coils inside the convective domain and barely visible but still dominant dipolar structure outside.

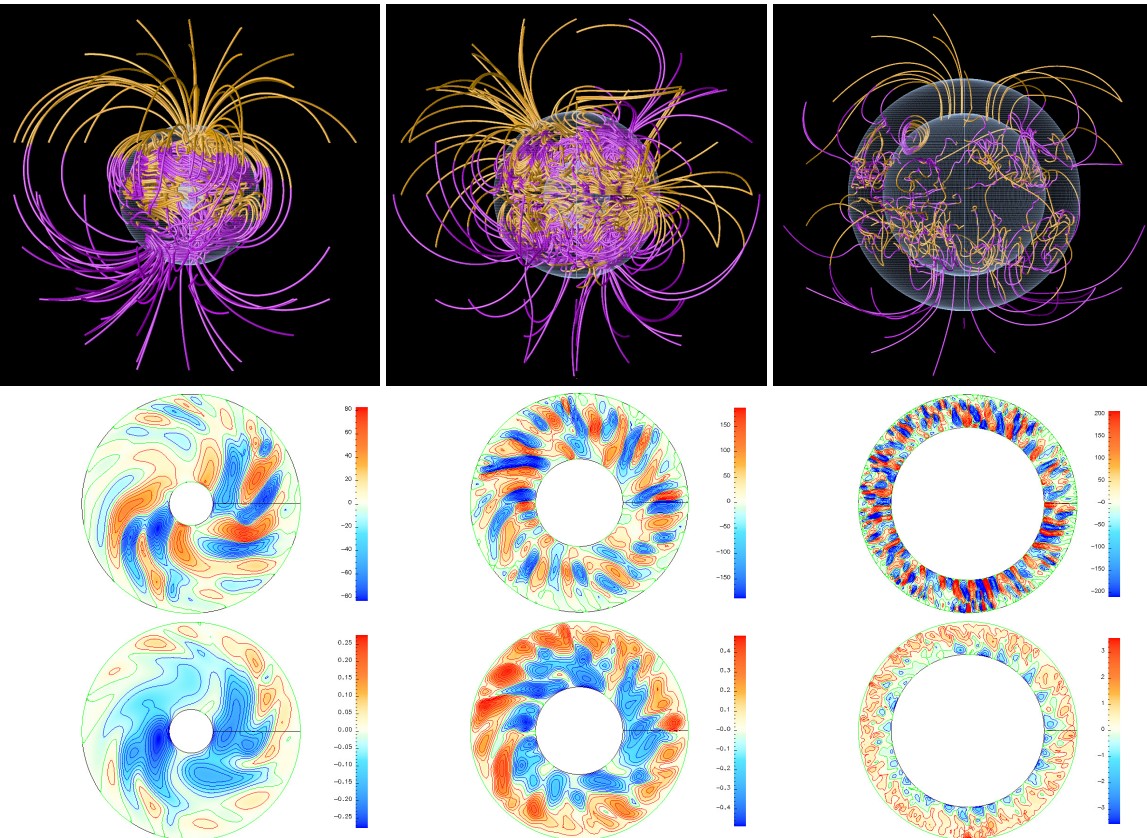

**Figure 4.** Snapshots of spatial structures of dynamo solutions with increasing shell thickness aspect ratio $\eta$ and with $R = 3.8 \times R_c$, $\tau = 2 \times 10^4$, $P = 0.75$ and $P_m = 1.5$. Three cases are shown as follows: $\eta = 0.2$, $R = 4,000,000$ (left column); $\eta = 0.4$, $R = 1,500,000$ (middle column); and $\eta = 0.7$, $R = 180,000$ (right column). Magnetic poloidal fieldlines are plotted in the top row, contours of the radial velocity $u_r$ in the equatorial plane are plotted in the middle row, and contours of the temperature perturbation $\Theta$ in the equatorial plane are plotted in the bottom row. (Colour online).

While typical, the spatial structures described in relation to Figure 4 are only snapshots of the three dynamo solutions at fixed moments in time. An illustration of the temporal behaviour exhibited in our dynamo simulations is shown in Figure 5. The main magnetic and kinetic energy density components of two distinct dynamo cases are plotted as functions of time, and the chaotic nature of the solutions is clearly visible. The time dependence of the time series consists of continual oscillations around the mean values of the respective densities with periods much shorter than the viscous diffusion time. Kinetic energy densities are displayed in the second row of the figure and show that the fluctuating components of motion dominate the flow with the fluctuating toroidal velocity being the strongest. The mean poloidal component of motion is negligible in both cases in agreement with the constraint of the Proudman–Taylor theorem on motions parallel to the axis of rotation. The mean toroidal component, representing differential rotation, appears to be weak in both cases plotted in Figure 5 more so in the case to the left marked **MD** for reasons we will discuss further below. The differential rotation, however, is known to be the component most strongly impaired in the presence of the magnetic field [34]. This leads us to a discussion of the features of the magnetic energy densities plotted in the first row of Figure 5. Here, the differences between the two cases illustrated are rather more pronounced. The total magnetic energy density of the case in Figure 5a is approximately six times larger than that in Figure 5d. More significant is the essential qualitative difference in the balance of magnetic energy components. The axisymmetric poloidal component $\overline{M}_p$ is dominant in the case shown in Figure 5a while it has a relatively small contribution in the case of Figure 5d. The axial dipole coefficient $H_1^0$ and the axial quadrupole coefficient $H_2^0$ in Figure 5c,f reveal that this

difference is due to the fact that the case to the left is dominated by a strong dipole and the case to the right is less strongly dipolar and the time series suggests the presence of magnetic field oscillations.

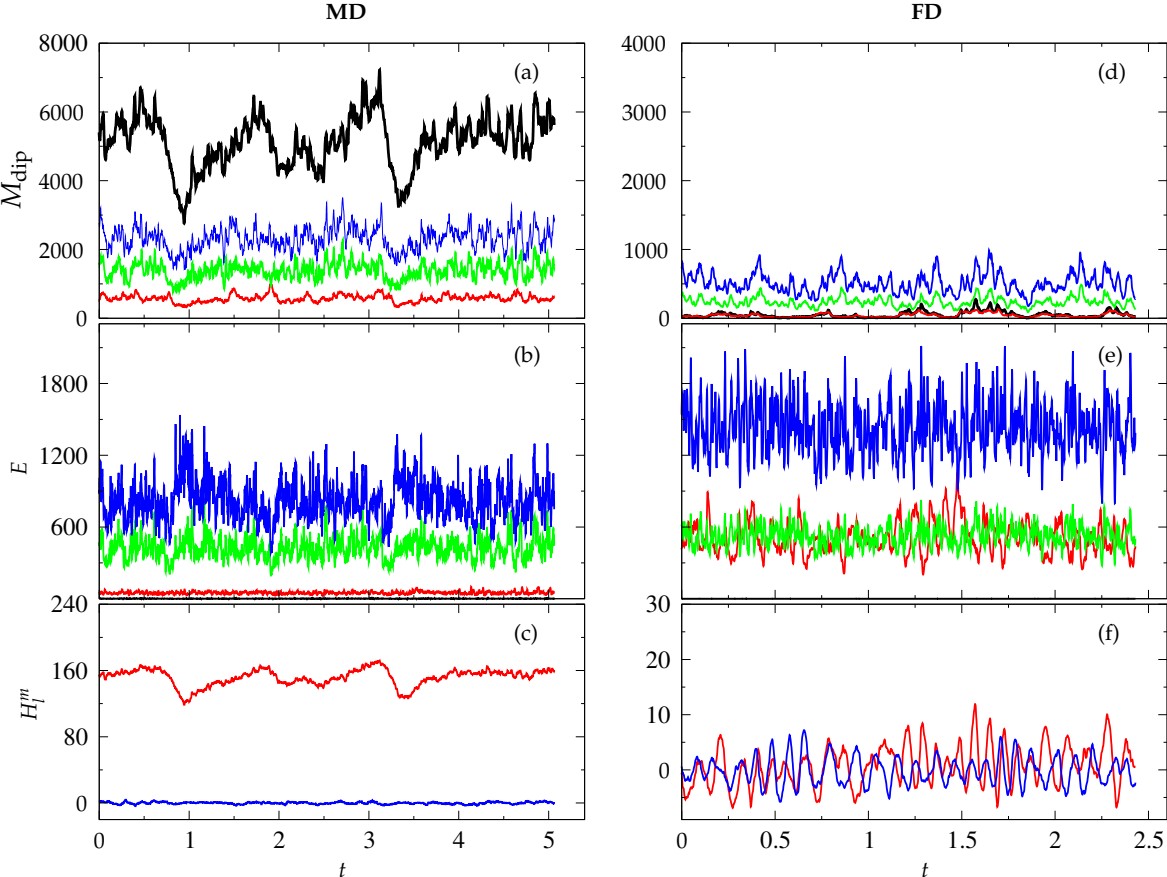

**Figure 5.** Chaotic dynamo attractors at identical parameter values—a Mean Dipolar (MD) dynamo (left column (**a**–**c**)) and a Fluctuating Dipolar (FD) dynamo (right column (**d**–**f**)) both at $\eta = 0.5$, $R = 8.2 \times 10^5$, $\tau = 2 \times 10^4$, $P = 0.75$ and $P_m = 1.5$. Panels (**a**,**d**) show time series of magnetic dipolar energy densities and panels (**b**,**e**) show kinetic energy densities. The component $\overline{X}_p$ is shown by solid black line, while $\overline{X}_t$, $\widetilde{X}_p$, and $\widetilde{X}_t$ are shown by red, green and blue lines, respectively. $X$ stands for either $M$ or $E$. Panels (**c**,**f**) show the axial dipolar $H_1^0$ and the axial quadrupolar $H_2^0$ coefficients at midshell $r = (r_i + r_o)/2$ by red and blue lines, respectively. Note the very different ordinate scales between panels (**a**,**c**,**d**,**f**). The ordinate scales of panels (**b**,**e**) are identical. (Colour online).

The solutions plotted in Figure 5a–f are examples of two types of dipolar dynamos that have been observed in numerical simulations [29,34,58,59], namely those with $\mathcal{D} > 1$ to which we will refer to as "Mean Dipolar" (**MD**) and those with $\mathcal{D} \leq 1$ that we will call "Fluctuating Dipolar" (**FD**). The typical spatial structures of the **MD** and **FD** dynamos are illustrated in Figure 6. The radial magnetic field plotted in the second column of Figure 6 shows the predominant dipolar symmetry of the dynamos, particularly clearly in the **MD** case where the north and the south hemispheres have opposite polarities entirely. The **FD** case displays a band of reversed polarity in a belt near the equator. In time this band propagates towards the poles and replaces the initial polarity leading to periodically occurring reversals. The stationary dipole of the **MD** case is stronger in intensity and inhibits differential rotation. This is confirmed by the profiles of the differential rotation plotted in the left part of the third column of Figure 6 that are markedly different. The **FD** case is characterised with a stronger geostrophic rotation largely aligned with the tangent cylinder while the mean zonal flow of the **MD** is weaker and exhibits a non-geostrophic rotation that is retrograde near the equator. The columnar convective structure of

the solutions remains similar in the **MD** and the **FD** case. Time-averaged kinetic and magnetic energy power spectra are shown in Figure 7.

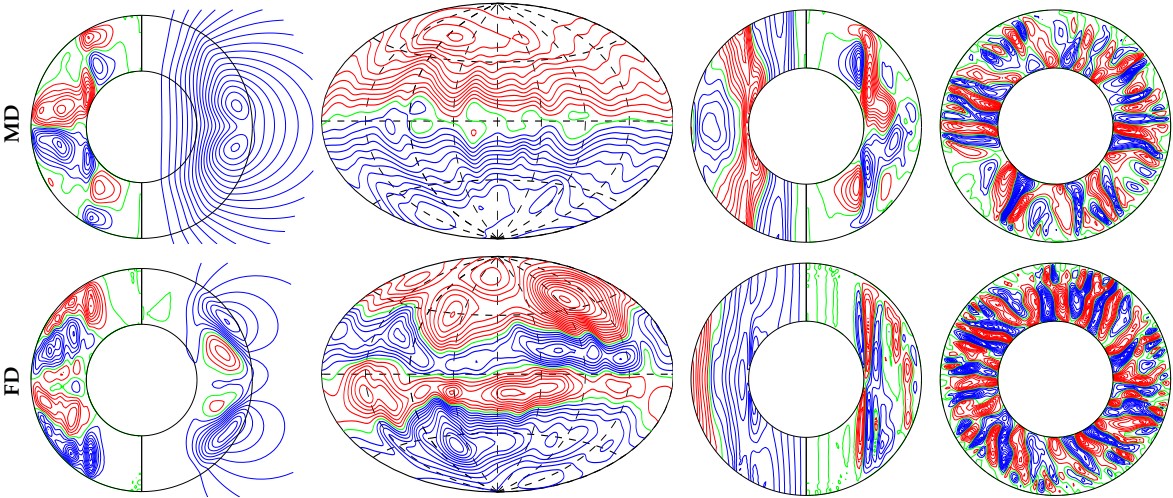

**Figure 6.** A **MD** (top row) and a **FD** (bottom row) dynamo solutions at $\eta = 0.5$, $R = 8.2 \times 10^5$, $\tau = 2 \times 10^4$, $P = 0.75$ and $P_m = 1.5$ corresponding to the cases shown in Figure 5. The first column shows meridional lines of constant $\overline{B_\varphi}$ in the left half and of $r \sin \theta \partial_\theta \overline{h} = const.$ in the right half. The second column shows lines of constant $B_r$ at $r = 1.675 r_o$. The third column shows meridional lines of constant $\overline{u}_\varphi$ in the left half and of $r \sin \theta \partial_\theta \overline{v}$ in the right half. The fourth column shows contours of the radial flow $u_r$ on the equatorial plane. Positive values are shown in red; negative values are shown in blue, and the zeroth contour line is shown in green. (Colour online).

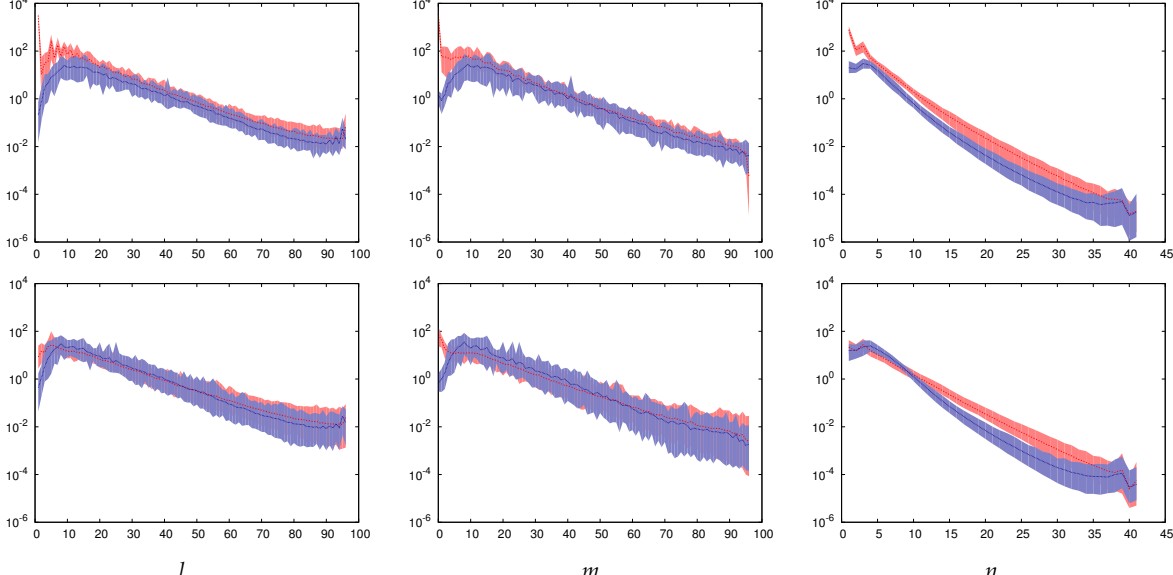

**Figure 7.** Typical power spectra of velocity (blue) and magnetic field (red). The top row shows an **MD** dynamo solution whereas the bottom row shows an **FD** dynamo solution both at $\eta = 0.4$, $R = 1,500,000$, $\tau = 2 \times 10^4$, $P = 0.75$ and $P_m = 1.5$. From left to right, power spectra as a function of the spherical harmonic degree $l$, order $m$, and Chebychev polynomial degree $n$ are shown, respectively. Lines represent the average spectra and shaded areas go from the minimum to the maximum values for each mode in the averaging period. A period of one viscous-diffusion time unit is used for the time-averaging period in both cases. (Colour online).

### 3.4. Bistability and General Effects of Shell Thickness Variation

One of the most remarkable features of **MD** and **FD** dynamos introduced above is that these two very distinct types can coexist at identical parameter values. Coexistence was first reported in [29]. Indeed, in each of the Figures 5–7, two different cases obtained at the same parameter values are shown. Within the parameter range of coexistence it is the initial conditions that determine which of the two chaotic attractors will be realised. Figure 8 shows the dipolarity ratio $\mathcal{D}$ as a function of the shell thickness aspect ratio $\eta$. Several observations can be made immediately. First, bistability only seems to occur for aspect ratios between $\eta = 0.25$ and $\eta = 0.6$ and both to the left and to the right of this interval **FD** dynamos are found. In contrast, alternating regimes appeared on each side of the hysteresis loop in previous studies [29,42] where continuation as a function of all remaining parameters $R$, $P$, $P_m$ and $\tau$ was performed. A further observation is that the **FD** dynamos have a decreasing dipolarity with increasing aspect ratio, that is, dipolarity seems to decrease with shell thickness. The **MD** dynamos, on the other hand, show little variation of dipolarity with aspect ratio but can still be separated into two groups, one for thin shells and another for thick shells. In this respect, it is apparent that thinner shells result in dynamos that are more dipole-dominated.

It is also interesting to note that there is a clear division between **MD** and **FD** dynamos also in the energy density space. Figure 9 shows a compilation of plots of magnetic energy density as a function of kinetic energy density. Dots represent instantaneous values; circles/triangles are mean values over time. The aspect ratio, $\eta$, increases from darker to lighter colours. Blue dots and circles represent simulations that started off as fluctuating dipolar dynamos whereas warm colours and greens represent simulations starting off as mean dipolar dynamos. Green symbols and dots represent simulations starting off as mean dipolar dynamos at $\eta = 0.6$ and $\eta = 0.7$ which were repeated starting from a higher magnetic energy and lower kinetic energy (triangles) relative to the original simulations (circles). Three regions can be clearly identified that correspond to simulations that finished as high and low dipolarity **MD** dynamos (regions I and II in Figure 9), and to simulations that finished as **FD** dynamos (region III in Figure 9). It is evident that dipolarity is preserved throughout the computations (most warm coloured dots and circles end up in region I and II; all blue dots and symbols end up in region III). The exception to this rule happens when the magnetic energy density of the initial **MD** condition is not big enough or its ration to the kinetic energy density is small (green circles). In this case the solutions drift to an **FD** state and remain there. If, on the other hand, the initial **MD** condition sees its magnetic energy density scaled up sufficiently, the solution will remain an **MD** dynamo (green dots and triangles).

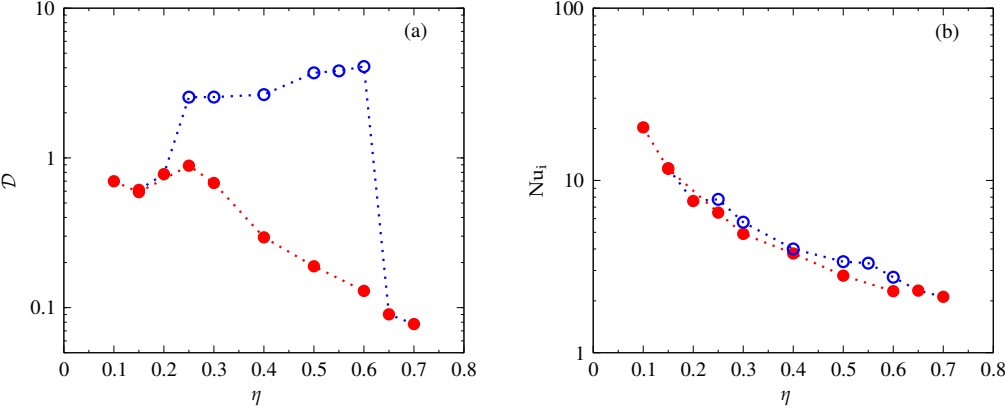

**Figure 8.** Bistability as a function of the shell thickness aspect ratio $\eta$. (**a**) The dipolarity ratio $\mathcal{D} = \overline{M}_p / \widetilde{M}_p$ and (**b**) the Nusselt number at $r = r_i$ in the cases $R = 3.8 \times R_c$, $\tau = 2 \times 10^4$, $P = 0.75$ and $P_m = 1.5$. Full red and empty blue circles indicate **FD** and **MD** dynamos, respectively. Red dotted lines and blue dotted lines connect dynamos that were started from **FD** and **MD** initial conditions, respectively. (Colour online).

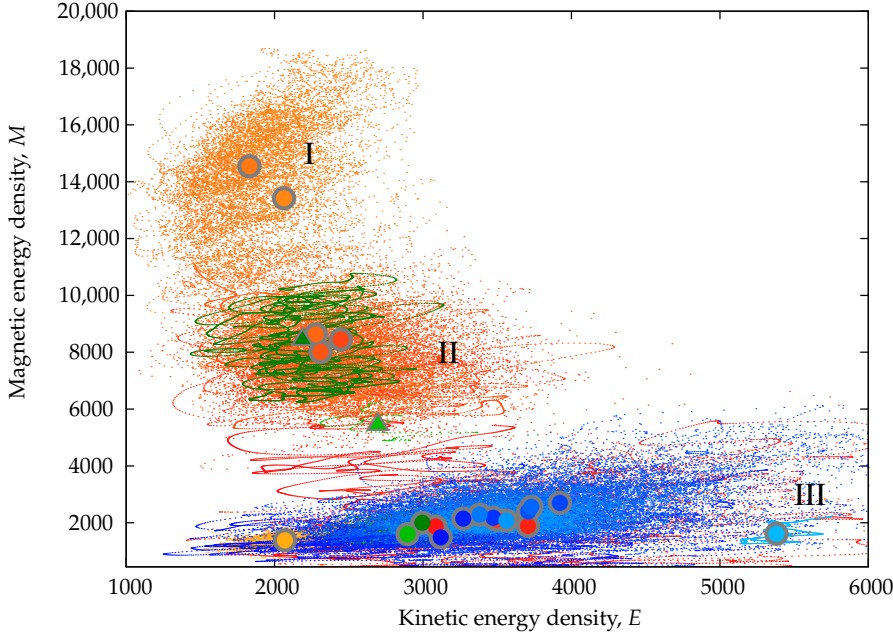

**Figure 9.** A "phase portrait" of magnetic vs. kinetic energy density values for dynamos with $R = 3.8 \times R_c$, $\tau = 2 \times 10^4$, $P = 0.75$ and $P_m = 1.5$. Dots are instantaneous values; large markers are time-averaged values. The aspect ratio $\eta$ increases from darker to lighter colours (blue to orange). Blue dots and points represent dynamos that were started from **FD** initial conditions. Warm colours and greens represent simulations that were started from **MD** initial conditions. Green symbols and dots represent simulations that were started as from **MD** initial conditions at $\eta = 0.6$ and $\eta = 0.7$ and that were repeated starting from a higher magnetic energy and lower kinetic energy (triangles) relative to the original simulations (circles). (Colour online).

### 3.5. The Cross-Helicity Effect

In order to model the effect of turbulence (or, at least, small-scale chaotic motion) on dynamo action, we consider a separation of scales. This approach is justified as dynamos tend to exhibit long-lasting large-scale structures (for example, the Earth's dipolar field) together with complex turbulent motions at smaller scales. We perform an averaging approach where, for the velocity field **u** and the magnetic field **b**, we write

$$\mathbf{u} = \mathbf{U} + \mathbf{u}', \tag{13a}$$

$$\mathbf{b} = \mathbf{B} + \mathbf{b}'. \tag{13b}$$

Capital letters represent large-scale components of each field, and will be referred to as the "mean" components within this and the following section. As described in the literature [15,16,20], there are several ways to perform this scale separation. Here, we perform the scale separation by assuming that the steady large-scale components of the flow and magnetic field can be identified with their respective time-averaged zonal components. The mean flow is then described as

$$\mathbf{U} = \langle \mathbf{u} \rangle = \frac{1}{2\pi\tau} \iint \mathbf{u} \, d\varphi \, dt, \tag{14}$$

for a suitable time scale $\tau$, and a similar expression can be constructed for the mean magnetic field. In principle, we can apply this separation of scales to all the main dynamical variables and all the model equations. Here, however, we only focus on the induction equation in order to gauge the effect of turbulent transport on the generation of the magnetic field through dynamo action.

Applying the above scale separation to the induction equation

$$\partial_t \mathbf{b} = \nabla \times (\mathbf{u} \times \mathbf{b}) + \lambda \nabla^2 \mathbf{b}, \tag{15}$$

where $\lambda$ is the magnetic diffusivity (note that Equation (16) is an alternative formulation of Equation (2e)), we find the induction equation for the mean magnetic field to be

$$\partial_t \mathbf{B} = \nabla \times (\mathbf{U} \times \mathbf{B}) + \nabla \times \mathbf{E}_M + \lambda \nabla^2 \mathbf{B}, \tag{16}$$

where the turbulent electromotive force, $\mathbf{E}_M$, is defined as

$$\mathbf{E}_M = \langle \mathbf{u}' \times \mathbf{b}' \rangle. \tag{17}$$

Through an application of the two-scale direct-interaction approximation (TSDIA) of inhomogeneous MHD turbulence (see [60] and references therein), the turbulent electromotive force can be written, in terms of mean variables, as

$$\mathbf{E}_M = \alpha \mathbf{B} - \beta \mathbf{J} + \gamma \mathbf{\Omega}. \tag{18}$$

Here, $\mathbf{J} = \nabla \times \mathbf{B}$ and $\mathbf{\Omega} = \nabla \times \mathbf{U}$. The coefficients $\alpha$, $\beta$ and $\gamma$ can be expressed in terms of the turbulent residual helicity, $H = \langle \mathbf{b}' \cdot \mathbf{j}' - \mathbf{u}' \cdot \boldsymbol{\omega}' \rangle$, the turbulent MHD energy, $K = \langle \mathbf{u}'^2 + \mathbf{b}'^2 \rangle / 2$, and the turbulent cross-helicity $W = \langle \mathbf{u}' \cdot \mathbf{b}' \rangle$, respectively [15,61]. Following [20], they are modelled as

$$\alpha = C_\alpha \tau \langle \mathbf{b}' \cdot \mathbf{j}' - \mathbf{u}' \cdot \boldsymbol{\omega}' \rangle = C_\alpha \tau H, \tag{19a}$$

$$\beta = C_\beta \tau \langle \mathbf{u}'^2 + \mathbf{b}'^2 \rangle = C_\beta \tau K, \tag{19b}$$

$$\gamma = C_\gamma \tau \langle \mathbf{u}' \cdot \mathbf{b}' \rangle = C_\gamma \tau W, \tag{19c}$$

with $C_\alpha$, $C_\beta$ and $C_\gamma$ being model constants. Here, $\tau$ is the characteristic time of turbulence, which is often expressed as

$$\tau = K/\epsilon, \tag{20}$$

with the dissipation rate of the turbulent MHD energy, $\epsilon$, defined by

$$\epsilon = \nu \left\langle \frac{\partial u_a'}{\partial x_b} \frac{\partial u_a'}{\partial x_b} \right\rangle + \lambda \left\langle \frac{\partial b_a'}{\partial x_b} \frac{\partial b_a'}{\partial x_b} \right\rangle. \tag{21}$$

Substituting (18) into the mean induction equation (16), we have

$$\partial_t \mathbf{B} = \nabla \times (\mathbf{U} \times \mathbf{B}) + \nabla \times (\alpha \mathbf{B} + \gamma \mathbf{\Omega}) - \nabla \times \left[ (\lambda + \beta) \nabla \times \mathbf{B} \right]. \tag{22}$$

Thus, in addition to the transport enhancement or structure destruction due to turbulence through the enhanced diffusion $\lambda + \beta$, there is also transport suppression or structure formation due to turbulence represented by the helicities $\alpha$ and $\gamma$ [60].

In the classical mean field theory of dynamos [10,15], the turbulent electromotive force is composed of the first two terms on the right-hand side of Equation (18), namely $\alpha \mathbf{B} - \beta \mathbf{J}$. Dynamos resulting from this model are known as "$\alpha$ dynamos", where the turbulent diffusion is balanced by an $\alpha$-effect. The properties of these terms have been discussed widely in the literature, and so we do not repeat this discussion here. Instead, let us now consider the final term on the right-hand side of Equation (18), $\gamma \mathbf{\Omega}$. Unlike the other terms describing the electromotive force, the mean variable in this term depends on the mean velocity and not the mean magnetic field. Yokoi [20] describes how a fluid element subject to a Coriolis-like force (a mean vorticity field) can contribute to the turbulent electromotive force through $\gamma$, a measure of the turbulent cross helicity. Dynamos in which the main balance is between $-\beta \mathbf{J}$ and $\gamma \mathbf{\Omega}$ are known as "cross-helicity dynamos", where the cross-helicity term replaces the $\alpha$-effect term in balancing the turbulent diffusion.

Cross-helicity dynamos have been studied much less than $\alpha$ dynamos, and this study represents an initial step in addressing this potentially important imbalance. In particular in Figure 10, we calculate all three contributions to the turbulent electromotive force in our dynamo simulations in order to determine their relative importance. These results are discussed below.

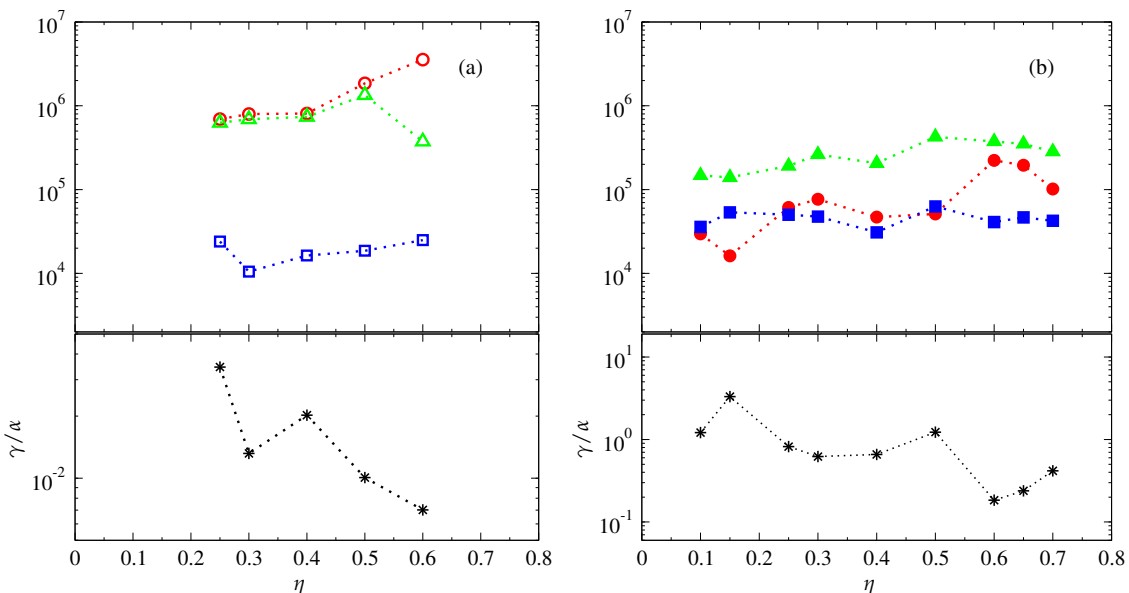

**Figure 10.** Magnitude of $\alpha$-, $\beta$-, and $\gamma$-effects with increasing shell thickness aspect ratio $\eta$ for dynamo solutions with $R = 3.8 \times R_c$, $\tau = 2 \times 10^4$, $P = 0.75$ and $P_m = 1.5$. The upper panels show root-mean squared time-averaged values of the $\alpha$-effect (red circles), $\beta$-effect (green triangles up) and $\gamma$-effect (blue squares). The lower panels show the ratio of $\gamma$-to $\alpha$-effects. Column (**a**) contains **MD** dynamo solutions (empty symbols) while column (**b**) contains **FD** dynamo solutions (full symbols) as shown in Figure 8. (Colour online).

### 3.6. Properties and Relative Importance of Cross-Helicity

The variation of the turbulent transport coefficients $\alpha$, $\beta$, and $\gamma$ as a function of shell thickness is displayed in Figure 10. For simplicity, in this initial investigation, we take $C_A\tau = 1$, where $A = \alpha$, $\beta$, or $\gamma$. Thus, the three effects are represented by the turbulent residual helicity $H$, the turbulent MHD energy $K$ and the turbulent cross-helicity $W$, respectively. For **MD** dynamo solutions, there is a clear disparity between the $\alpha$- and $\beta$-effects, and the $\gamma$-effect. The $\gamma$-effect is, for the range of $\eta$ considered, about two orders of magnitude smaller than the other effects. Thus, across a wide range of shell thickness aspect ratios, **MD** dynamos can be considered to be operating predominantly as $\alpha$ dynamos. In contrast, for **FD** dynamo solutions, a different picture emerges. Across the range of $\eta$ considered, the $\alpha$- and $\gamma$-effects are of a similar magnitude. Thus, both these effects are potentially important in balancing the $\beta$-effect. Therefore, **FD** dynamo solutions represent a "mixture" of an $\alpha$ dynamo and a cross-helicity dynamo.

Figure 11 displays $z$-projections of the azimuthally-averaged components of the electromotive force. For the **MD** dynamo solutions, shown in (a), the $\gamma$-effect follows an antisymmetric pattern about the equator, just like the other effects. This behaviour is expected from the pseudoscalar nature of $\gamma$ and the symmetry of magnetic fields in **MD** dynamos [20]. For **FD** dynamo solutions, such as those displayed in (b), the components of the electromotive force no longer exhibit antisymmetry about the equator. This behaviour is, in part, due to the more complex spatial structure of the magnetic fields of **FD** dynamos compared to **MD** dynamos. This feature, combined with generally weaker magnetic field strengths and different flow profiles (see Figures 5 and 6, for example), results in the $\alpha$-effect being weaker for **FD** dynamos. Thus, both the $\alpha$- and $\gamma$-effects become of comparable importance in sustaining dynamo action.

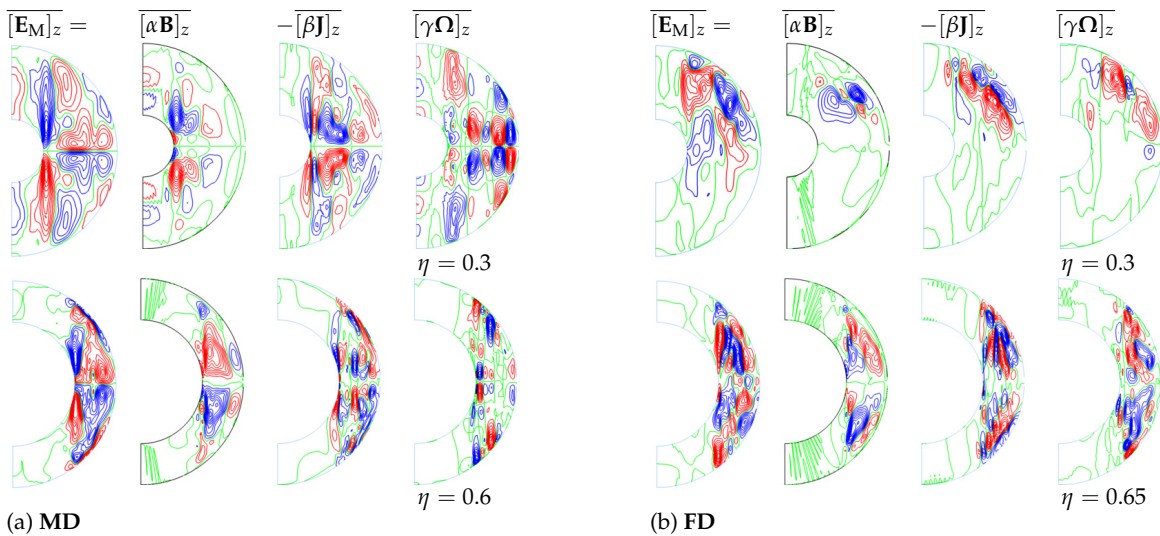

**Figure 11.** Spacial structures of the azimuthally-averaged $z$-component of the electromotive force $\mathbf{E}_M$ and its $\alpha$-, $\beta$- and $\gamma$-effect constituents as given by Equation (18). Four dipolar dynamo solutions are plotted as follows. (**a**) **MD** dynamo solutions with $\eta = 0.3$, $P = 0.75$, $\tau = 2 \times 10^4$, $R = 2{,}500{,}000$, $P_m = 1.5$ (**top row**) and $\eta = 0.6$, $P = 0.75$, $\tau = 2 \times 10^4$, $R = 410{,}000$, $P_m = 1.5$ (**bottom row**). (**b**) **FD** dynamo solutions with $\eta = 0.3$, $P = 0.75$, $\tau = 2 \times 10^4$, $R = 2{,}500{,}000$, $P_m = 1.5$ (**top row**) and $\eta = 0.65$, $P = 0.75$, $\tau = 2 \times 10^4$, $R = 300{,}000$, $P_m = 1.5$ (**bottom row**). In each column contour lines of the quantities denoted at the column heading are plotted with positive contours shown in red, negative contours shown in blue, and the zeroth contour shown in green. (Colour online).

## 4. Summary and Discussion

Rotating thermal convection is ubiquitous within the interiors and the atmospheres of celestial bodies. These fluid regions usually contain plasmas or metallic components so vigorous convection drives large-scale electric currents and generates the self-sustained magnetic fields characteristic of these cosmic objects. In this article the relative importance of two main mechanisms for magnetic field generation and amplification is assessed, namely the helicity- and the cross-helicity effects of mean-field dynamo theory. The motivation for this study was to test the hypothesis that the turbulent helicity effect, also known as the $\alpha$-effect, is more important in the case of the geodynamo, while the cross-helicity effect, also known as the $\gamma$-effect, is more significant in the case of the solar global dynamo, due to differences between the shell aspect ratio of the solar convection zone and that of Earth's inner core. The following novel results are reported in the article.

(a) Critical parameter values for onset of convection determined numerically as functions of the shell radius ratio, $\eta$.

(b) Bistability and coexistence of two distinct dynamo attractors found as a function of the shell radius ratio, $\eta$.

(c) Spatial distributions and time-averaged values of turbulent helicity and cross-helicity EMF effects obtained (1) for both types of dynamo attractors, as well as (2) as functions of the shell radius ratio, $\eta$.

Further details and a discussion of these results follows.

To assess $\alpha$- and $\gamma$- electromotive effects, we performed, and report here, an extensive suite of over 40 direct numerical simulations of self-sustained dynamo action driven by thermal convection in rotating spherical fluid shells, where the shell thickness aspect ratio $\eta$ is varied at fixed values of the other parameters. The simulations are based on the Boussinesq approximation of the governing nonlinear magnetohydrodynamic equations with stress-free velocity boundary conditions. While the use of fully compressible equations is desirable, it is not feasible for global dynamo simulations.

Indeed, the fully compressible MHD equations allow sound wave solutions with periods many orders of magnitude shorter than the convective turnover time and the magnetic diffusion timescales that are of primary interest. The Boussinesq approximation is justified and generally used for modelling convection in Earth's inner core where density variation between the inner–outer core boundary and the core mantle boundary is small [13,14,38,47]. The density contrast between the bottom ($\rho_i$) and the top ($\rho_o$) of the Solar convection zone is five orders of magnitude giving a density scale number of $\log(\rho_i/\rho_o) \approx 12$ [62], and the anelastic approximation is more appropriate and commonly used in global solar convection models, for example, [11,33,63]. However, anelastic and Boussinesq simulations show many similarities [63], with Boussinesq models able to mimic solar periodicity and active longitude phenomena [25,42]. Thus, in this work the Boussinesq approximation is used for uniformity across various shell radius ratios and to focus on the effects of shell thickness in isolation from effects of density stratification.

Coexistence of distinct chaotic dynamo states has been reported to occur in this problem in terms of certain governing parameters in [29,31]. In this study, we establish that two essentially different nonlinear dynamo attractors coexist also for an extensive range of shell thickness aspect ratios $\eta \in [0.25, 0.6]$. Since this is precisely the range of values where most celestial dynamos operate, this result is significant as it demonstrates that field morphologies may be dependent on the initial state of a dynamo. We proceed to discuss in detail the contrasting properties characterising the coexisting dynamo regimes (mean-field dipolar (**MD**) dynamos and fluctuating dipolar (**FD**) dynamos) including differences in temporal behaviour and spatial structures of both the magnetic field and rotating thermal convection. We find that the relative importance of the electromotive dynamo effects is different in the cases of mean-field dipolar dynamos and fluctuating dipolar dynamos. The helicity $\alpha$-effect and the cross-helicity $\gamma$-effect are comparable in intensity in the case of fluctuating dipolar dynamos and their ratio does not vary significantly with shell thickness. In contrast, in the case of mean-field dipolar dynamos the helicity $\alpha$-effect dominates by approximately two orders of magnitude and becomes even stronger with decreasing shell thickness. Our results, therefore, indicate that both dynamo mechanisms are important for solar global magnetic field generation as the solar dynamo is of a fluctuating dipolar type. Our results also indicate that the cross-helicity effect may be important in understanding dynamo mechanisms in stellar dynamos. The latter may also be of fluctuating dipolar type and markedly different from the solar dynamo, for example, having large-scale magnetic structures being dominant in only one hemisphere [64]. Since the geodynamo is of a mean-field dipolar type, the helicity effect appears, indeed, to be more significant in this case and our results show this effect will become even stronger as the inner solid core grows in size by iron freezing. Simulations of the geodynamo with nucleation and growth of the inner core have been recently reported by Driscoll [65] and Landeau et al. [66]. These authors find that pre-inner core nucleation dynamos exhibit weak thermal convection, low magnetic intensity and non-dipolar field morphology, while post-inner core nucleation and with increasing inner core size their solutions have stronger axial dipole morphology. Our results similarly demonstrate that **FD** and multipolar dynamos occur when the value of the shell radius ratio $\eta$ is smaller than 0.25. However, our **FD** solutions exhibit vigorous convection and can be described as strong-field dynamos even though they are of lower magnetic field intensity than corresponding **MD** dynamos. A further discrepancy is that for $\eta > 0.25$ we find that **MD** and **FD** dynamos coexist. These discrepancies can be attributed to significant differences in thermal and velocity boundary conditions between our model and the models of [65,66]. Most importantly, the governing parameters values in [65,66] are controlled by thermochemical evolution models and vary with inner core size (age), while in our study all parameter values apart from $\eta$ are kept fixed.

It will be of interest to revisit the analysis of helicity and cross-helicity effects using the more general anelastic approximation of the governing equations. Further, there are many questions that remain to be answered on how the dynamic balance between the components of the electromotive force affects different aspects of dynamo action, including how to switch between **MD** and **FD** dynamos.

**Author Contributions:** Conceptualization, R.D.S.; methodology, R.D.S.; software, R.D.S.; validation, L.S.; formal analysis, R.D.S. and D.M.; investigation, L.S. and P.G.; resources, R.D.S.; data curation, L.S. and P.G.; writing—original draft preparation, R.D.S.; writing—review and editing, R.D.S. and D.M.; visualization, L.S., R.D.S. and P.G.; supervision, R.D.S.; funding acquisition, R.S. All authors have read and agreed to the published version of the manuscript.

**Funding:** This research was funded by the Leverhulme Trust grant number RPG-2012-600.

**Acknowledgments:** Numerical simulations were carried out in part at the DiRAC Data Centric system at Durham University, operated by the Institute for Computational Cosmology on behalf of the STFC DiRAC HPC Facility (www.dirac.ac.uk). This equipment was funded by BIS National E-infrastructure capital grant ST/K00042X/1, STFC capital grants ST/H008519/1 and ST/K00087X/1, STFC DiRAC Operations grant ST/K003267/1 and Durham University. DiRAC is part of the National E-Infrastructure.

**Conflicts of Interest:** The authors declare no conflict of interest.

## Abbreviations

The following abbreviations are used in this manuscript:

**MD**    Mean Dipolar Dynamo
**FD**    Fluctuating Dipolar Dynamo

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
