# Peer review of "Effects of Shell Thickness on Cross-Helicity Generation in Convection-Driven Spherical Dynamos"

_fluids, doi:10.3390/fluids5040245_

Round 1

Reviewer 1 Report

The authors deal with the actual topic "The relative importance of the helicity and cross-helicity electromotive dynamo effects for self-sustained magnetic field generation by chaotic thermal convection in rotating spherical shells". They combine direct numerical simulation with a mean-field magnetohydrodynamic approach. The numerical model is based on a hydromagnetic dynamo driven by thermal convection, in which they study the characteristics of the generated magnetic field in dependence on the size of the inner core. Their results brought the coexistence of two distinct branches of dynamo solutions - a mean-field dipolar regime and a fluctuating dipolar regime. These two modes are further analyzed using the mean-field MHD approximation to point out the effect of the helicity α-effect and the cross-helicity γ-effect on magnetic field generation.

The article is neatly processed. Before to be accepted for publication,
I have a few comments:

  1. this analysis is beneficial for the study of geodynamo and solar dynamos. For numerical modelling, they use the Boussinesq approximation, which is a suitable approximation for the geodynamo. How applicable is it for the study of the solar dynamo, compared to the anelastic approximation, or considering the full compressible fluid? In Discussion, the authors set a goal for the following research - the extension of this task using an anelastic approximation. However, in the model description, it would be appropriate to state how suitable the Boussinesq approximation is for geodynamo and how much for the solar dynamo.
  2. in the chapter Results and Discussion, it would be appropriate to compare your results with findings presented in

Driscoll P.E., 2016. Simulating 2 Ga of geodynamo history, Geophys. Res. Lett., 43, 1–8. DOI: 10.1002 / 2016GL68858

and

Landeau M., Aubert J. and Olson P., 2017. The signature of inner-core nucleation on the geodynamo, Earth Planet. Sci. Lett., 465, 193–204

Besides, I have some minor comments:

Pg 10, L249: filedline -> fieldline

Reviewer 2 Report

An extremely high 39% similarity is recorded at this manuscript that make it inappropriate for publication. Especially, since most of the plagiarism concerns another article of some of the authors, that makes under skepticism the novelty on the manuscript.

Round 2

Reviewer 2 Report

Authors improved the manuscript and justify its novelty according to our suggestion, so it can be accepted for publication in the Fluids journal.